



# Passive seismic recording of cryoseisms in Adventdalen, Svalbard

Rowan Romeyn[1,2], Alfred Hanssen[1,2], Bent Ole Ruud[2,3], Helene Meling Stemland[2,3], Tor Arne Johansen[2,3,4]

[1]Department of Geosciences, University of Tromsø – The Arctic University of Norway, 9037 Tromsø, Norway
[2]Research Centre for Arctic Petroleum Exploration (ARCEx)
[3]Department of Earth Science, University of Bergen, 5007 Bergen, Norway
[4]The University Centre in Svalbard (UNIS), 9171 Longyearbyen, Norway

*Correspondence to*: Rowan Romeyn (rowan.romeyn@uit.no)

**Abstract.** A series of transient seismic events were discovered in passive seismic recordings from 2D geophone arrays deployed at a frost polygon site in Adventdalen, Svalbard. These events contain a high proportion of surface wave energy and produce high-quality dispersion images through an innovative source localisation approach, based on apparent offset resorting and inter-trace delay minimisation, followed by cross-correlation beamforming dispersion imaging. The dispersion images are highly analogous to surface wave studies of pavements and display a complex multimodal dispersion pattern. Supported by

theoretical modelling based on a highly simplified arrangement of horizontal layers, we infer that a ~3.5-4.5 m thick, stiff, high-velocity layer overlies a ~30 m thick layer that is significantly softer and slower at our study site. Based on previous studies we link the upper layer with syngenetic ground-ice formed in aeolian sediments, while the underlying layer is linked to epigenetic permafrost in marine-deltaic sediments containing unfrozen saline pore water. Comparing events from spring and autumn shows that temporal variation can be resolved via passive seismic monitoring. The transient seismic events that

we record occur during periods of rapidly changing air temperature. This correlation along with the spatial clustering along the elevated river terrace in a known frost polygon, ice-wedge area and the high proportion of surface wave energy constitutes the primary evidence for us to interpret these events as frost quakes, a class of cryoseism. In this study we have proved the concept of passive seismic monitoring of permafrost in Adventdalen, Svalbard.

## 1    Introduction

Permafrost is defined as ground that remains at or below 0°C for at least two consecutive years (French, 2017). On Svalbard, an archipelago located in the climatic polar tundra zone (Kottek et al., 2006), at least 90% of the land surface area not covered by glaciers is underlain by laterally continuous permafrost (Christiansen et al., 2010; Humlum et al., 2003). A seasonally active layer, where freezing/thawing occurs each winter/summer, extends from the surface to a depth of 0.8-1.2 m (Christiansen et al., 2010) and overlies the permafrost. However, the purely thermal definition of permafrost means that the mechanical

properties can vary widely depending on the actual ground-ice content. The ground-ice content varies spatially according to sediment texture, organic content, moisture availability and sediment accumulation rate (Gilbert et al., 2016; Kanevskiy et al.,



2011; O'Neill and Burn, 2012). Because of the significant impact of ice content on mechanical strength, seismic velocities are a relatively sensitive tool to study the subsurface distribution of ground-ice (Dou et al., 2017; Johansen et al., 2003).

The thermal dynamics of this permafrost environment lead to an interesting phenomenon called cryoseisms, sometimes referred to as frost quakes. Cryoseisms are produced by the sudden cracking of frozen material at the Earth's surface (Battaglia et al., 2016). They are typically observed in conjunction with abrupt drops in air and ground temperature below the freezing point, in the absence of an insulating snow layer and in areas where high water saturation is present in the ground (Barosh, 2000; Battaglia et al., 2016; Matsuoka et al., 2018; Nikonov, 2010). When the surface temperature drops well below 0°C the

frozen permeable soil expands, increasing the stress on its surroundings, which can eventually lead to explosive pressure release and tensional fracturing (Barosh, 2000; Battaglia et al., 2016). Seismic waves from these events decay rapidly with distance from the point of rupture, but have been felt at distances of several hundred meters to several kilometres and are usually accompanied by cracking or booming noises, resembling falling trees, gunshots or underground thunder (Leung et al., 2017; Nikonov, 2010). The zero focal depth of cryoseisms means that, relative to tectonic earthquakes, a larger proportion of

the energy is distributed in the form of surface waves (Barosh, 2000).

Methods based on the analysis of surface waves are used extensively in engineering fields, such as the non-destructive testing of structures or assessment of the mechanical properties of soils relating to their use as a foundation for built structures (Chillara and Lissenden, 2015; Park et al., 1999; Park et al., 2007; Rose, 2004). In a typical soil profile, the shear velocity and stiffness

of the ground increase gradually with depth due to compaction leading to a simple wavefield dominated by fundamental mode Rayleigh waves (Foti et al., 2018). Mechanical properties of the ground are then estimated relatively simply from the geometrical dispersion of the recorded wavefield, i.e., the measured pattern of phase velocity as a function of frequency, where lower frequencies interact with the ground to greater depths than higher frequencies.

By contrast, in permafrost environments, the surface layer freezes solidly during the winter leading to an increase in the shear modulus of the upper layer (Johansen et al., 2003) and an inverse shear velocity with depth profile. This is similar to the case of pavement in civil engineering studies, where a thin, relatively stiff, high-velocity layer at the surface overlies softer and slower ground materials beneath. The high-velocity surface layer acts as a waveguide, permitting the excitation of higher-order wave modes and the wavefield becomes significantly more complicated due to the large number of simultaneously propagating

wave modes (Foti et al., 2018; Ryden and Lowe, 2004). Such engineering applications have furthermore driven the development of wave propagation models capable of modelling these complicated wavefields. For example, the ground may be represented by a horizontally layered medium with partial wave balance at the interfaces, for which the dispersion spectrum and stress-displacement field is readily calculated using the global matrix method (Lowe, 1995).

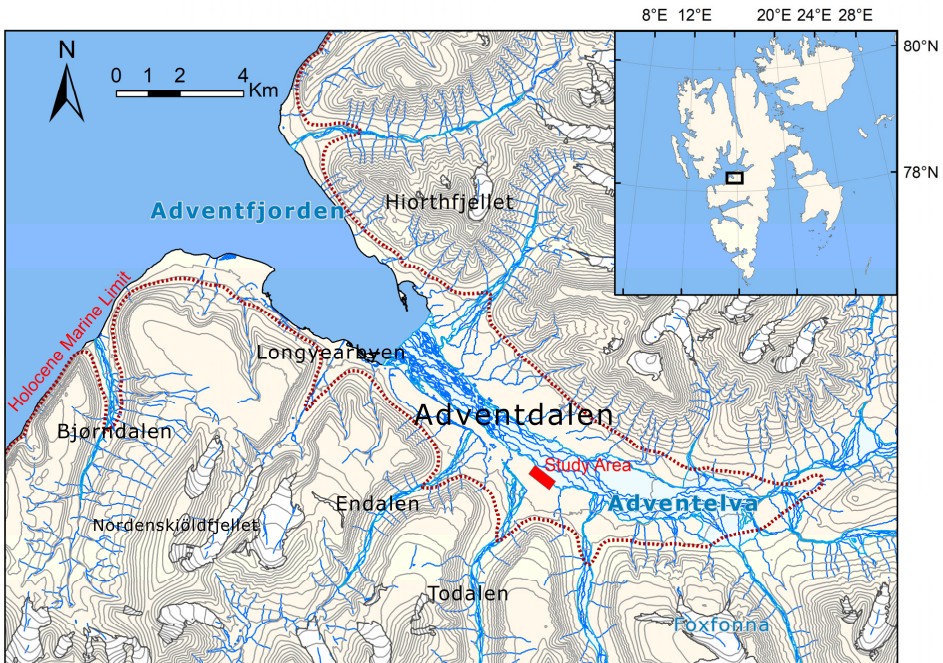

**Figure 1: 1:100 000 scale map showing the location of the study area (red box) in Adventdalen. Inset map illustrates the location with respect to the Svalbard archipelago. The Holocene marine limit (red dashed line) is drawn according to Lønne (2005). Map data © Norwegian Polar Institute (npolar.no).**

## 2    Study area and seismic acquisition

Our study site is located in Adventdalen, near the main settlement Longyearbyen on the island Spitsbergen within the Svalbard
archipelago in the high Arctic as shown in Figure 1. The climate of this area, as recorded at Longyearbyen airport, is
characterised by low mean annual precipitation of 192 mm, and mean annual air temperature of -5.1°C for the period 1990-
2004, rising to -2.6°C during the period 2005-2017. The maritime setting and alternating influence of low pressure systems
from the south and polar high pressure systems means that rapid temperature swings are common during winter from above
0°C down to -20 or -30°C, while summer temperatures are more stable in the range of 5-8°C (Matsuoka et al., 2018). Snow
cover in the study area is typically shallow, due to the strong winds that blow along the valley and varies according to local
topography with 0-0.1 m over local ridges and 0.3-0.4 m in troughs. Positive temperature and snowmelt events also occur
sporadically through the freezing season (October to May) and subsequently result in the formation of a thin ice-cover. Both
shallow snow-cover and thin ice-cover permit efficient ground cooling throughout the freezing season (Matsuoka et al., 2018).



Adventdalen has continuous permafrost extending down to ~100 m depth (Humlum et al., 2003), but the highest ground-ice content is restricted to the uppermost ~4 m in the loess-covered river terraces that bound the relatively flat, braided Adventelva river plain. The geological setting of the study site is illustrated in Figure 2. The formation of permafrost in Adventdalen began around 3 ka concurrent with the subaerial exposure and onset of aeolian sedimentation on valley-side alluvial terraces (Gilbert et al., 2018). Sediment cores studied by Cable et al. (2018) and Gilbert et al. (2018) show a consistent pattern of an increased

ground-ice content over a ~4 m thick interval (decreasing towards the coast), beneath a ~1 m thick active layer. This interval was observed consistently in cores retrieved from alluvial and loess deposits and is interpreted as ice rich syngenetic permafrost (Cable et al., 2018; Gilbert et al., 2018).

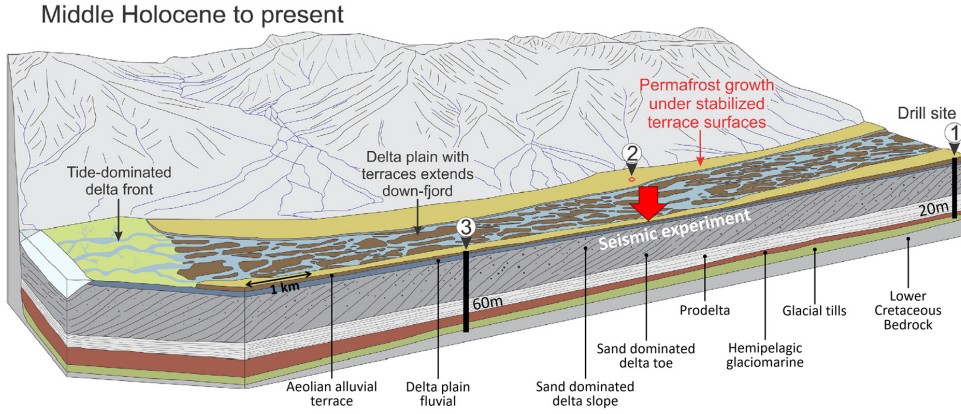

**Figure 2: Geological model of Adventdalen, Svalbard modified from Gilbert et al. (2018), the numbered sites mark the coring locations studied by Gilbert et al. (2018) and the red arrow marks the position of the seismic experiments described in this study.**

The underlying interval consists of marine-influenced deltaic sediments into which permafrost has grown epigenetically and where ground-ice content remains low (Gilbert et al., 2018). Unfrozen permafrost has been mapped in Adventdalen below the

Holocene marine limit (which includes the entire study area as illustrated in Figure 1) using nuclear magnetic resonance and controlled source audio-magnetotelluric data (Keating et al., 2018). This is again related to the presence of saline pore-water in these marine-influenced deltaic sediments that causes the pore-water to remain at least partially unfrozen despite sub-zero temperatures.

Ice wedge polygons, one of the most recognizable landforms in permafrost environments (Christiansen et al., 2016), are present at the investigated site in Adventdalen. They form when freezing winter temperatures cause the ground to contract and crack under stress (Lachenbruch, 1962). Water later infiltrates the cracks and refreezes as thin ice veins that extend down into the



permafrost. These veins have lower tensile strength compared to the surrounding ground (Lachenbruch, 1962; Mackay, 1984), so subsequent freeze induced cracking occurs preferentially along this plane of weakness. The repeated cracking, infilling and

refreezing causes the ice wedges to grow laterally, forcing the displaced ground upwards and resulting in a series of ridges in a polygonal arrangement that are the surface hallmark of the phenomenon (Christiansen et al., 2016; Lachenbruch, 1962).

Sudden ground accelerations corresponding to cryoseismic events have previously been observed at our study site in Adventdalen. Matsuoka et al. (2018) monitored three ice-wedge troughs within an area of polygonal patterned ground at the

site, using a combination of extensometers, accelerometers and breaking cables connected to timing devices. Their study, which extended over the 12 year period 2005-2017, provides a valuable overview of the seasonality and correspondence between ground motion and environmental parameters. The related study of O'Neill and Christiansen (2018), further details the accelerometer results. Ice-wedge cracking was typically registered in late winter, when the top of the permafrost cooled to around -10°C, resulting in large accelerations of 5 g to more than 100 g. However, O'Neill and Christiansen (2018) also report

smaller magnitude accelerations throughout the freezing season, typically in conjunction with rapid surface cooling, that are thought to be caused by the initiation of cracks within the active layer or the horizontal and vertical propagation of existing ice wedge cracks. Given the timing of the field campaigns for the present study, during spring and autumn of 2019, we expect it is rather the latter category of events that have been recorded.

In the present study ground motion was recorded using geophones deployed in 2D arrays (the geometry of which is discussed in sect. 3.3.1). During the spring field campaign, groups of 8 gimballed vertical-component geophones connected in series (geophone type Sensor SM-4/B 14 Hz, 0.7 damping and spurious frequency of 190 Hz) were deployed on the snow surface at each receiver location. During deployment for the autumn field campaign, geophones were embedded into the unfrozen ground surface as an assortment of spike geophones (Sercel SG-10 10Hz) connected in series in strings of 4 geophones and 3C

geophones (DT-Solo 3C with z-element HP301V-10Hz) where only the vertical channel was used. Both of these geophone types have damping of 0.7, giving a flat response above the natural frequency up to the spurious frequency of 240 Hz. Data was recorded for defined time intervals as will be discussed in sect. 4.1, mandated primarily by battery considerations.

## 3   Methods

In this study, we present a methodology to isolate transient seismic signals in passive seismic recordings from two-dimensional

vertical component geophone arrays. These transient signals contain surface wave energy with relatively high signal-to-noise ratio. We implement a novel method to localise the unknown source position of these signals based on the 2D receiver array geometry and subsequently recover dispersion spectra using a cross-correlation beamforming technique. In order to infer subsurface physical properties we generate theoretical dispersion curves using the global matrix method (Lowe, 1995) based on idealized horizontally layered media models. The forward model is manually tuned to achieve best fit with the





experimentally observed dispersion spectra. Similar experiments were conducted over two field campaigns in the spring and autumn in order to investigate temporal variation in permafrost mechanical strength.

### 3.1 Isolation of microseismic events in passive records

Our passive seismic recordings contain a significant amount of non-surface wave energy, including wind noise and air waves.
As a result we find it more effective to isolate and analyse specific transient microseismic events rather than attempting to
recover the Green's function from ambient noise cross correlations as, e.g., Sergeant et al. (2020) have done for passive recordings on glaciers. The periodic microseismic signals are isolated from background random noise based on permutation entropy, a nonlinear statistical measure of randomness in a time series (Bandt and Pompe, 2002), that produces local minima for coherent signals embedded in noise. We use the implementation of Unakafova and Keller (2013) using ordinal patterns of third order extracted over successive samples and a sliding window size of 200 samples. We then apply a peak-finding
algorithm to identify local minima in permutation entropy that meet peak prominence criteria defined by thresholds of peak value, height and width. An example of event detection is shown in Figure 3 for real, noisy data recorded at the study site.

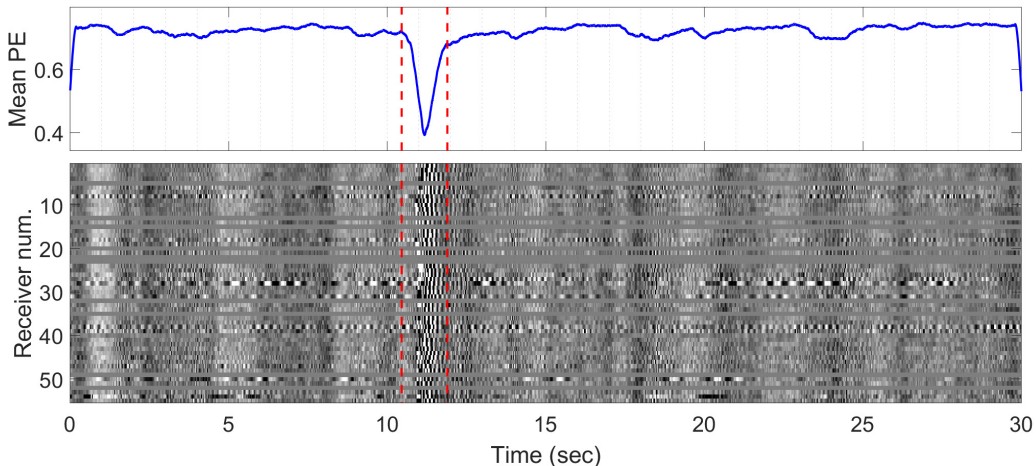

**Figure 3: Detection of transient event based on mean permutation entropy (PE), a metric that peaks towards minima when coherent amplitudes are recorded across the array of receivers. Red dashed lines mark the temporal extent of the extracted event that is**
**shown in greater detail in Figure 4. This event was recorded 30-Mar-2019 at 18:06.**

The isolated microseismic events can be subsequently processed in a similar way to active source experiments, i.e. using well established multichannel analysis of surface waves (MASW) methodologies, with modification due to the unknown source position.




### 3.2 MASW dispersion imaging with unknown source position

Several processing methods for multichannel analysis of surface waves (MASW) are possible depending on the acquisition setup. One of the most well-known is the 1D phase shift method of Park et al. (1998), that is straightforward to apply for line arrays of receivers and inline sources at known offsets. The dispersion image is built by scanning over frequency ($\omega$) and phase velocity ($v$) according to,

$$E_{1D}(\omega, v) = \left| \sum_{i=1}^{N} e^{j\phi_i} R_i(\omega) \right|, \tag{1}$$

where $R_i(\omega)$ is the Fourier transform of the $i$-th trace $r_i(t)$ of $N$ recorded traces and $\phi_i = \omega x_i / v$ is the corresponding phase shift at known source-receiver offset $x_i$. On the other hand, processing of passively recorded microseismic events is complicated by the fact that the source position is unknown. We employed a 2D receiver array to allow us to localise the unknown passive seismic sources. Again, there are several possibilities to process the 2D array data. For example, Park et al. (2004) describe an azimuth scanning technique assuming far-field sources and utilizing the plane-wave projection principle. The dispersion image is formed according to,

$$E_{2D}(\omega, v, \theta) = \left| \sum_{i=1}^{N} e^{j\phi_x} e^{j\phi_y} R_i(x, y, \omega) \right|, \tag{2}$$

where $R_i(x, y, \omega)$ is the Fourier transform of the $i$-th trace $r_i(x, y, t)$, located at position $(x, y)$, of $N$ recorded traces and $\phi_x = -\omega x \cos\theta / v$ and $\phi_y = -\omega y \sin\theta / v$ are the phase shifts corresponding to the $x$ and $y$ components of the phase velocity, where $\theta$ denotes the source azimuth. We implement this approach by computing, for each combination of velocity and frequency, the azimuth that maximizes the spectral magnitude. The source azimuth is thus estimated by calculating the modal azimuth across all frequencies and velocities and the dispersion spectrum can be formed by fixing the azimuth and reiterating over frequency and velocity axes. The key drawback of forming the dispersion image according to Eq. (2) is that the source should be distant enough that the plane wave assumption is valid. In Figure 4 we demonstrate that our experimental data is not consistent with the far-field source approximation, since moving the source position closer to the array simplifies the structure of the apparent offset sorted gather. This observation led us to develop an alternative processing approach.

### 3.3 Two-step MASW approach

We propose a two-step processing approach where the first step involves locating the unknown source position, permitting the dispersion image to be formed in the second step. Even the simplest 1D phase shift method from Eq. (1) will give superior results to Eq. (2) if a nearby source can be reliably located. Our approach is based on the idea that if the source position was


known, the seismic traces from the 2D array could be arranged by their apparent offset from the source to form a shot gather that resembles the simple case of a line array with an inline source at known offset. Figure 4 shows that when we determine the source azimuth using Eq. (2) and re-sort the recorded traces by offset to a distant source lying along this azimuth we

observe that the gather begins to resemble the response of a line array. However, when we shift the assumed source position closer to the array, the offset sorted gather resembles the simple linear array response even more closely. Thus, by formalising a metric that encodes the resemblance of an apparent offset sorted gather to a line array response, we can obtain a useful tool for source localisation that leverages the 2D array geometry. This further avoids the problem of picking specific P- and S-phase arrivals, a traditional seismological method for source localisation, since these arrivals are difficult to detect reliably in

our experimental data.

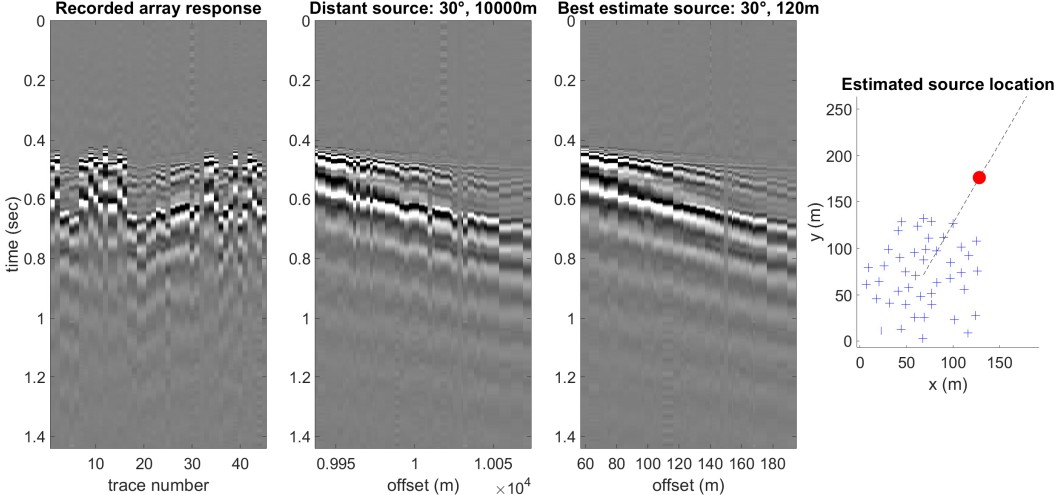

**Figure 4 – Field recording of a transient event, whose detection is highlighted in Figure 3, demonstrates that re-sorting traces by apparent offset to a distant source produces a gather with poorer coherence than re-sorting by offset to the best estimate source**
**range at 120 m. Blue crosses denote geophones, while the red circle marks the best estimate source position.**

### 3.3.1  Step 1 - Source localisation

The source is located by re-sorting the seismic traces by offset to a series of test positions. The validity of a given test position is assessed by summing the magnitude of the delays between neighbouring traces in the re-sorted gather. The delays between

neighbouring traces are estimated based on the lag that gives maximal value to the normalised cross-correlation between the two signals. The delays are estimated on a relatively narrowband filtered copy of the seismic (10-15-30-45 Hz Ormsby filter), to minimise the influence of random noise and dispersion. After sorting, the source position is selected as the test position that

produces the smallest magnitude sum of adjacent trace delays, corresponding to the simplest and most coherent offset sorted gather. This simple approach works well in practice and is fast enough to allow a relatively large number of test positions to

be evaluated. However, the most coherent offset sorted gather may correspond to unphysical negative velocities, indicating the true source azimuth lies 180° from the selected azimuth. This situation is identified and corrected by comparing summed frequency-wavenumber (FK) transform magnitudes corresponding to negative $k$-space to those of positive $k$-space. If negative $k$-space produces a larger sum than positive $k$-space, a negative dip corresponding to unphysical negative velocities exists and we rotate the selected source azimuth 180°.


In Figure 5 we show that the source azimuths estimated using this technique are consistent with those estimated by picking maxima in azimuth scans using Eq. (2), but have the additional benefit that the source range is also estimated.

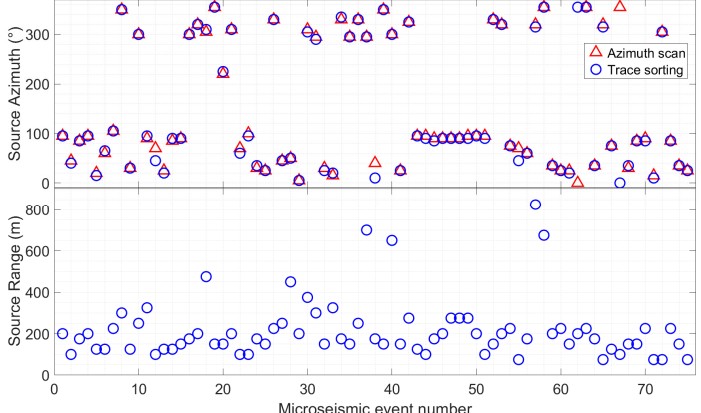

**Figure 5 – Comparison of source localisation using the Azimuth Scanning approach, i.e., Eq. (2) (red triangles) and the new approach based on trace re-sorting and delay minimisation (blue circles) which also allows range estimation. Azimuths are given as compass bearings.**

The reliability of the proposed method of source localisation was tested using synthetic gathers corresponding to a set of known source azimuths and ranges. We ran a 1D noise-free forward model (that accounts for 3D wavefield divergence) using the

OASES package for seismo-acoustic propagation in horizontally stratified waveguides, which employs a wavenumber-integration solution method (Schmidt and Jensen, 1985). We model the wavefield using the layer properties corresponding to the spring field conditions detailed in Table 1. We then specify the receiver positions as deployed in the field and form the synthetic gathers by selecting traces with appropriate offset from the 1D pre-calculated wavefield. The range and azimuth errors produced when attempting to recover the known source positions using the proposed source localisation approach are

illustrated in Figure 6. The proposed method demonstrates an excellent ability to recover the direction to the source, regardless

of azimuth, although uncertainties relating to the true receiver positions in the field are also important to consider (see sect. 3.3.3). We also observe reliable estimation of source range within a radius of ~500 m from the array centre, beyond which we observe an increasing tendency to underestimate the source range. Further tests with a range of different array geometries indicate that the array aperture is the dominant factor controlling the maximum source range that can be estimated reliably.


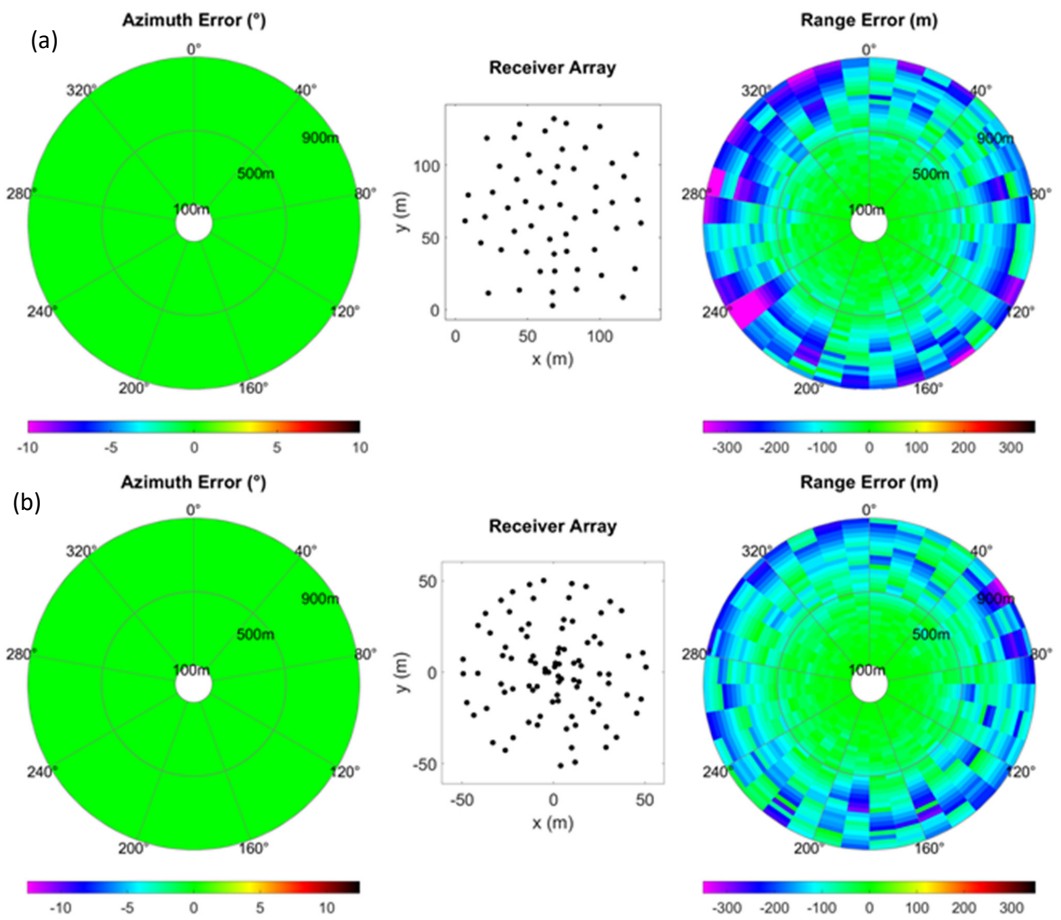

Figure 6: Predicted source localization azimuth and range errors based on forward modelling with known source positions and receiver geometry corresponding to (a) spring and (b) autumn field campaigns.




### 3.3.2    Step 2 - Dispersion imaging

Once the seismic traces from the 2D array have been sorted by the apparent offset to the localised source, we are left with a
gather that resembles a linear receiver array and an inline source, as seen in Figure 4. At this point, the dispersion image may
simply be formed by applying Eq. (1), i.e., the 1D phase shift method of Park et al. (1998). However, more advanced processing

methodologies have emerged over time and we observe significantly improved dispersion imaging using the cross-correlation
beamforming approach of Le Feuvre et al. (2015), as demonstrated in Figure 7. This approach utilizes the cross-correlations
between all possible pairs of receivers, rather than the recorded traces themselves, to increase the effective spatial sampling of
the array and thereby reduce aliasing and increase signal-to-noise ratio. In an adaption of this approach, we form the dispersion
image, $D(\omega, v)$, a function of frequency ($\omega$) and phase velocity ($v$), according to the following equation using the source

position, $r_s(x, y)$, estimated as described in Sect. 3.3.1:

$$D(\omega, v) = \left| \sum_{j=1}^{N_R-1} \sum_{k=j+1}^{N_R} \delta(\omega, v, r_s, r_j, r_k) \right|, \tag{3}$$

with $N_R$ the number of receivers and $r_j(x, y)$, $r_k(x, y)$ denoting the positions of the receivers for the cross-correlation

pair. Furthermore:

$$\delta(\omega, v, r_s, r_j, r_k) = \begin{cases} \tilde{C}_{jk}(\omega) e^{i\omega \frac{\|r_s - r_k\| - \|r_s - r_j\|}{v}} & if \ \|r_s - r_j\| \ \leq \ \|r_s - r_k\|, \\ \tilde{C}_{kj}(\omega) e^{i\omega \frac{\|r_s - r_j\| - \|r_s - r_k\|}{v}} & if \ \|r_s - r_j\| \ > \ \|r_s - r_k\|. \end{cases} \tag{4}$$

Here, the causal cross-correlations $\tilde{C}_{jk}$ and $\tilde{C}_{kj}$ between the receivers located at $r_j$ and $r_k$ are selected according to the direction

of propagation, determined by comparing the two source-to-receiver distances, while the propagation distance is given by the
difference between the two. In practice, we do not compute the cross correlation $\tilde{C}_{kj}$ directly, but instead use the equivalent
time-reversed acausal part (negative time delays) of $\tilde{C}_{jk}$ (Le Feuvre et al., 2015). It is important that the seismic traces are pre-
whitened prior to computing the cross correlations, as whitening effectively removes the autocorrelation of the signals that can
blur the cross-correlation (El-Gohary and McNames, 2007). We find that a simple first-order backward differencing scheme

is an effective method to whiten the recorded traces. We furthermore find it convenient to normalise the frequency response
of the dispersion spectrum so that the maximum amplitude along a given frequency is unity.

It should also be noted that it is possible to localise the source by searching for source positions that produce dispersion images
with maximum amplitude, as the correct source location is expected to produce the most coherent high-amplitude dispersion

modes (Le Feuvre et al., 2015). This approach was tested in the present study. However, we find this method very slow and



inefficient compared to the trace resorting approach we implement, even though maximising dispersion image magnitude does appear a valid approach for locating an unknown source.

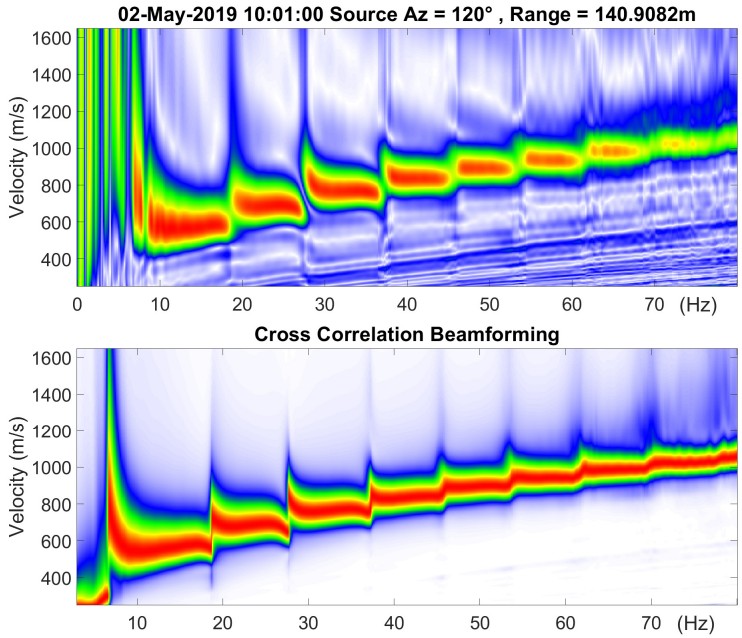

**Figure 7: Dispersion spectra using 1D phase shift method (top) and Le Feuvre et al. (2015) cross correlation beamforming (bottom). Source position was localised using delay minimisation approach in both cases and colour scales are linear.**

### 3.3.3    Influence of receiver position errors

During the field campaigns, we recorded GPS positions at the recording nodes that the receivers are connected to, rather than at the receivers themselves. The positions of the receivers were subsequently assigned based on an approximate dead reckoning approach and therefore have a degree of uncertainty associated with them. We estimate that this positional uncertainty lies in the range of ~2-3 m. The impact of receiver position errors was investigated using an OASES 1D seismo-acoustic propagation model (Schmidt and Jensen, 1985), for the horizontally stratified waveguide corresponding to the spring field conditions detailed in Table 1. We extracted a reference gather assuming a source range of 200 m, an azimuth of 100° and receiver geometry corresponding to the spring field campaign. We then ran 1000 iterations adding flat-spectrum random perturbations to the receiver positions. By setting the maximum amplitude of the perturbation, we effectively define a circle with radius corresponding to this amplitude, where there is equal probability that the receiver is positioned at any given point within this circle. We then measure the impact of these perturbations on both source localisation and dispersion spectra. The dispersion





spectrum error ($\varepsilon^2$) is given by the sum of squared differences for a given noise-perturbed trial $S_{test}$, compared to a noise-free reference spectrum $S_{ref}$ over $n$ frequencies ($\omega$) and $m$ velocities ($v$):

$$\varepsilon^2 = \sum_{i=1}^{n} \sum_{j=1}^{m} \Big( S_{test}(\omega_i, v_j) - S_{ref}(\omega_i, v_j) \Big)^2 .$$

(5)


In Figure 8 we show that positional errors up to 4-5 m in radius have a very minor impact on source azimuth estimation and dispersion spectra in the frequency range of interest, i.e., up to 100 Hz. Range estimation is somewhat more sensitive (but less important to dispersion spectrum quality) and we see a general trend that the source range tends to be overestimated rather than underestimated under the influence of receiver position uncertainty. This indicates that the estimated positional

uncertainty for the field campaigns (~2-3 m) should not significantly affect our experimental results, although we may expect some minor overestimation of source range. As the positional error magnitude increases further, Figure 8 demonstrates that the source localisation becomes progressively more imprecise, while Figure 9 shows that the maximum frequency imaged coherently in the dispersion spectra progressively decreases. We can formalise this trend by observing the relation that coherent dispersion spectra are recovered for wavelengths of approximately 2-3 times the maximum positional error magnitude.


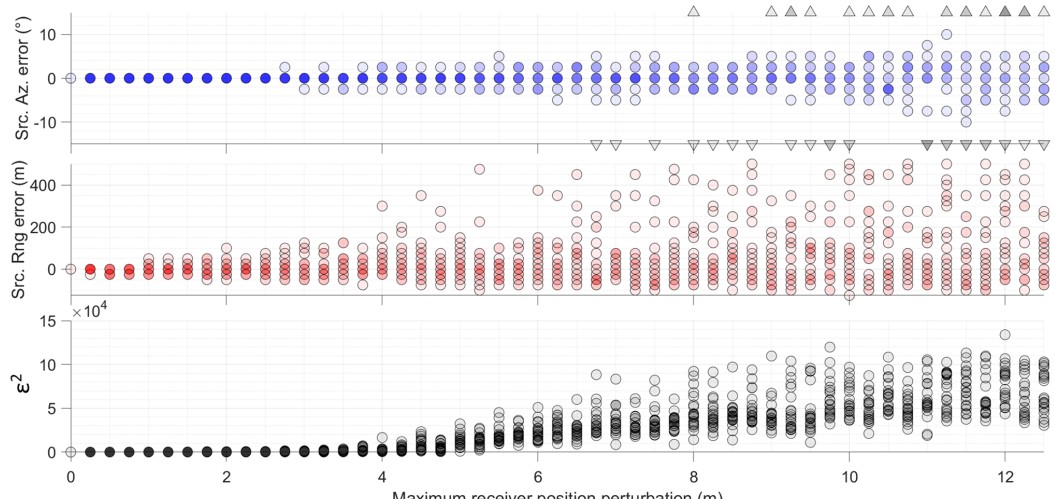

**Figure 8: Summary of results of 1000 modelling iterations with random white noise perturbation to receiver positions. Grey triangles denote outliers, i.e., iterations that result in large azimuth errors outside the plotted range. Higher colour density denotes overlap, i.e. multiple iterations with the same result. $\varepsilon^2$ denotes dispersion spectrum error Eq. (5).**




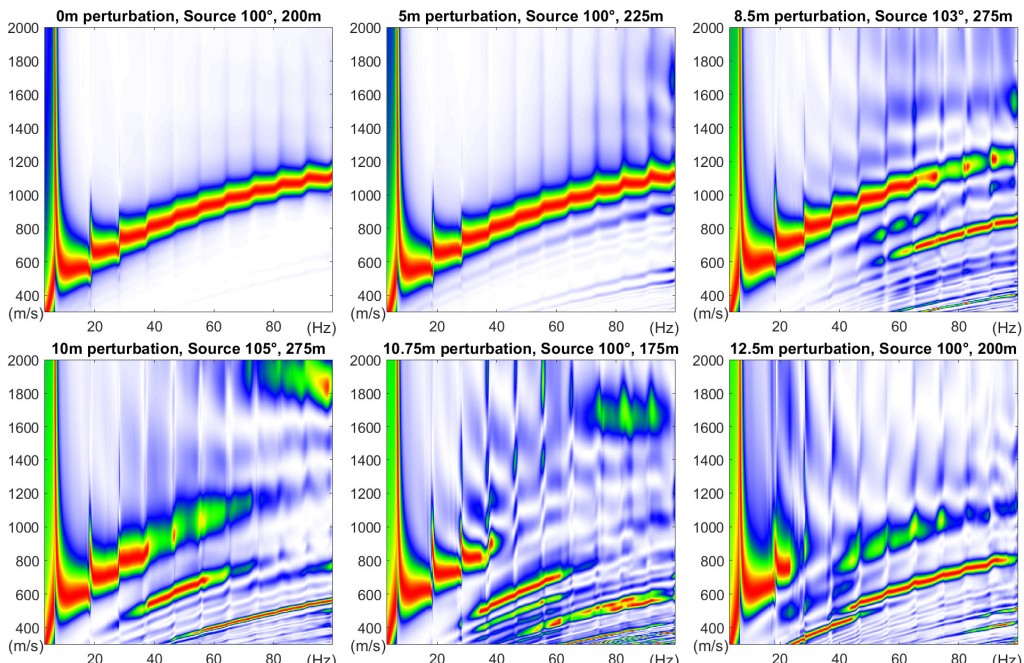

**Figure 9: Dispersion spectra calculated from forward model with receiver positions perturbed by white random noise of known magnitude illustrating how the maximum frequency that is coherently imaged decreases with increasing uncertainty in receiver position. Colour scale is linear.**

### 3.4 Theoretical dispersion curve modelling

The global matrix method was introduced by Knopoff (1964), further elaborated by Lowe (1995) and again by Ryden and Lowe (2004). It involves the assembly of a system matrix $\boldsymbol{S}$ that describes the interaction of displacement and stress fields across interfaces between horizontal layers described by a series of interface matrices. The propagating wavemodes are characterised by combinations of frequency ($\omega$) and wavenumber ($k$) that satisfy all boundary conditions such that the determinant vanishes:

$$f(\omega, k) = \det[\boldsymbol{S}] = 0 \tag{6}$$

We abstain from providing a full derivation, but give a specific case study that illustrates our implementation and may serve as a simple practical reference point for the reader interested in further exploration of the method. The layer models used in this study contain two discrete layers ($i = 2, 3$) bounded by an infinite vacuum half-space above ($i = 1$) and a solid half-space below ($i = 4$), giving the system matrix the following form:





$$S = \begin{bmatrix} D_{1b}^- & -D_{2t} & \\ & D_{2b} & -D_{3t} \\ & & D_{3b} & -D_{4t}^+ \end{bmatrix},$$ (7)

where the matrices describing the top interfaces $D_t$ and the bottom interfaces $D_b$ for each layer are given by the following expressions (noting that the minus superscript denotes taking only the upward travelling partial waves given by columns two and four, while the plus superscript denotes selecting only the downward travelling partial waves given by columns one and three):

$$[D_{it}] = \begin{bmatrix} k & kg_\alpha & C_\beta & -C_\beta g_\beta \\ C_\alpha & -C_\alpha g_\alpha & -k & -kg_\beta \\ i\rho_i B & i\rho_i B g_\alpha & -2i\rho_i k B^2 C_\beta & 2i\rho_i k B^2 C_\beta g_\beta \\ 2i\rho_i k B^2 C_\alpha & -2i\rho_i k B^2 C_\alpha g_\alpha & i\rho_i B & i\rho_i B g_\beta \end{bmatrix},$$ (8)

$$[D_{ib}] = \begin{bmatrix} kg_\alpha & k & C_\beta g_\beta & -C_\beta \\ C_\alpha g_\alpha & -C_\alpha & -kg_\beta & -k \\ i\rho_i B g_\alpha & i\rho_i B & -2i\rho_i k B^2 C_\beta g_\beta & 2i\rho_i k B^2 C_\beta \\ 2i\rho_i k B^2 C_\alpha g_\alpha & -2i\rho_i k B^2 C_\alpha & i\rho_i B g_\beta & i\rho_i B \end{bmatrix},$$ (9)

with:

$$g_\alpha = e^{iC_\alpha h_i}, \quad g_\beta = e^{iC_\beta h_i} \quad ,$$ (10)

$$C_\alpha = \left(\frac{\omega^2}{\alpha_i^2} - k^2\right)^{\frac{1}{2}}, \quad C_\beta = \left(\frac{\omega^2}{\beta_i^2} - k^2\right)^{\frac{1}{2}} \quad ,$$ (11)

$$B = \omega^2 - 2\beta_i^2 k^2,$$ (12)

and the physical properties of the system enter as:

$h_i$ = thickness of layer $i$, zero for the half-spaces

$\rho_i$ = density of layer $i$, set to zero for the upper vacuum half-space

$\alpha_i$ = bulk compressional velocity of layer $i$, arbitrary non-zero value for upper vacuum half-space

$\beta_i$ = bulk shear velocity of layer $i$, arbitrary non-zero value for upper vacuum half-space.

We are also interested in the magnitude of displacement at the ground surface (top interface of layer $i$=2) for the different wave modes so that we may predict which are most likely to be excited and subsequently recorded in the field. To this end, we proceed by assuming the amplitudes of the incoming waves in the two half spaces, setting a unitary amplitude entering the





system at the top ($\{A_1^+\} = 1$) and zero amplitude entering the system from below ($\{A_4^-\} = 0$), allowing us to specify the right-hand side of the following system and solve for the unknown interface amplitudes using a least squares approach:

$$\begin{bmatrix} \boldsymbol{D}_{1b}^- & -\boldsymbol{D}_{2t} & & \\ & \boldsymbol{D}_{2b} & -\boldsymbol{D}_{3t} & \\ & & \boldsymbol{D}_{3b} & -\boldsymbol{D}_{4t}^+ \end{bmatrix} \begin{Bmatrix} \{A_1^-\} \\ \{A_2\} \\ \{A_3\} \\ \{A_4^+\} \end{Bmatrix} = \begin{bmatrix} \boldsymbol{D}_{1b}^- & & & \\ & & & -\boldsymbol{D}_{4t}^+ \end{bmatrix} \begin{Bmatrix} \{A_1^+\} \\ 0 \\ 0 \\ \{A_4^-\} \end{Bmatrix}. \tag{13}$$

Here we substitute the system matrix $\boldsymbol{S}$ for the combinations of frequency and wavenumber that correspond to the propagating
345 wavemodes. The vectors of amplitudes are arranged in the following way:

$$\{A\} = \begin{Bmatrix} A_{(L+)} \\ A_{(L-)} \\ A_{(S+)} \\ A_{(S-)} \end{Bmatrix}, \quad \{A^+\} = \begin{Bmatrix} A_{(L+)} \\ A_{(S+)} \end{Bmatrix}, \quad \{A^-\} = \begin{Bmatrix} A_{(L-)} \\ A_{(S-)} \end{Bmatrix} \quad , \tag{14}$$

where $L$ and $S$ denote longitudinal and shear waves respectively, while – and + symbols denote upward and downward travelling partial waves. Having solved for the unknown amplitudes in Eq. (13) we then calculate the displacements and stresses at the ground surface according to the following equation,

$$350 \quad \begin{Bmatrix} u_x \\ u_z \\ \sigma_{zz} \\ \sigma_{xz} \end{Bmatrix}_{2t} = [\boldsymbol{D}_{2t}]\{A_2\} \quad , \tag{15}$$

where $u_x$ and $u_z$ denote the complex valued in-plane and vertical displacements, while $\sigma_{zz}$ and $\sigma_{xz}$ denote the complex valued vertical and lateral stresses and the calculation is made at the top interface of layer 2.

In Figure 10, we show an example of the dispersion curves produced by this approach for the spring layer properties listed in Table 1. Since we measured the vertical component of ground motion in the field, we plot the magnitude of the vertical
355 displacement ($u_z$) as an indicator of the relative likelihood of exciting and subsequently recording specific wave modes. We also highlight, for a given frequency, the wavemode giving the largest displacement at the surface that is considered most likely to dominate the ground response and contribute to the apparent dispersion curve produced by the superposition of multiple modes observed in experimental data.
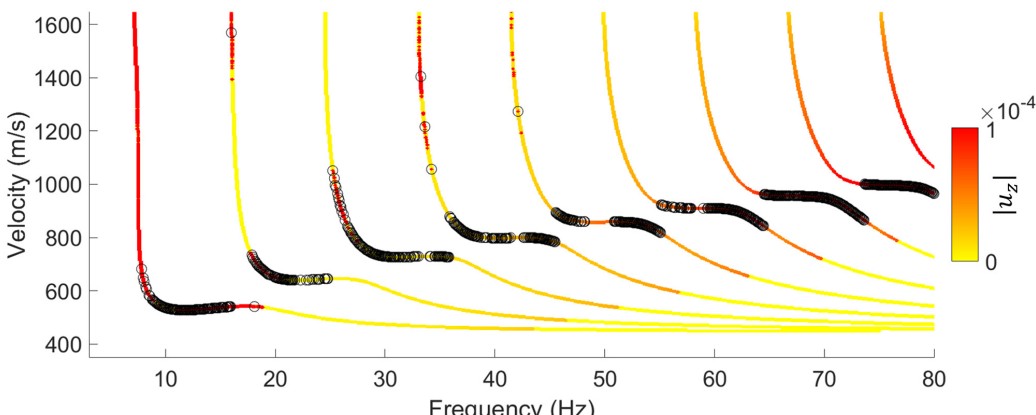

**Figure 10: Example of theoretical dispersion curves coloured by magnitude of vertical displacement at the ground surface. Black circles denote the wavemode with largest displacement for a particular frequency.**

### 3.4.1 Numerical root finding method

To recover the dispersion curves it is necessary to find the roots of the system matrix, Eq. (6). In this study, we assume a half-space with higher velocity than the overlying layers, which reflects the presence of compacted sediments and bedrock at depth in Adventdalen. The implication of this choice is that the propagating surface wave modes do not leak energy into the half-space and are much simpler to search for numerically. The permitted surface wave modes are given by combinations of frequency and phase velocity (or wavenumber) that minimize the determinant of the system matrix. We localize these minima by conducting a simple 2D numerical search over a regular grid of frequency and real wavenumber (the imaginary part of the wavenumber is zeroed since we consider only non-leaky modes). The local minima in the matrix sampling the determinant of the system matrix are found by morphological image processing techniques, rather than the more traditional method of curve tracing using bisection algorithms favoured by, e.g., Lowe (1995). Specifically, we use the "*imregionalmax*" routine in Matlab on the negative of the determinant matrix, applying a series of linear connectivity kernels for detection of local maxima ridges. We then mask out non-dispersive body waves that appear as horizontal ridges in frequency-phase velocity space and use morphological closing operations to fill in the gaps that this creates. We further apply skeletonization to the binary image of dispersion curves. The non-zero elements of the binary matrix then define the combinations of frequency and velocity that represent the dispersive wave modes and that are subsequently used to calculate surface wave amplitudes with Eq. (13).





## 4    Results and discussion

### 4.1    Interpretation of cryoseisms


A series of transient seismic events were isolated from passive seismic recordings of 2D geophone arrays deployed at our study site in Adventdalen. The estimated source positions for these events are shown in Figure 11, while their temporal occurrence is illustrated in Figure 12. The events cluster primarily around frost polygons along the raised river-terrace. A small

number of events fall just beyond the raised riverbank and plot within the Adventelva river valley. While these events may be correctly located, we also cannot rule out the possibility that they occurred on the raised terrace and the source range has been overestimated as discussed in sect. 3.3.3. Figure 12 shows that the transient events were all recorded during periods of rapidly changing air temperature as recorded at a nearby weather station. This observation together with the spatial clustering around frost polygons and on the raised river terrace, which is known to have high ground-ice content within the upper ~4 m (Gilbert

et al., 2018), lead us to infer that these events are most likely cryoseisms, or frost quakes. The fact that these events consist dominantly of surface wave energy is also consistent with a shallow source and previous descriptions of cryoseisms (Barosh, 2000). Similarly, the fact that the source range for all of the recorded events was in the order of hundreds of meters is also consistent with the distance over which previously observed cryoseisms have propagated (Leung et al., 2017; Nikonov, 2010) and we are unaware of any other likely seismic sources within this range. Other possible seismic sources such as an operational

coal mine, Gruve 7, that conducts blasting operations lies ~5 km SE, road traffic along the road ~650-850 m S-SW or snowmobile traffic along the Adventelva river valley N-E of the study site do not explain the spatial distribution and character of the recorded events. Known examples of snowmobile and vehicle traffic contain strong air wave arrivals with non-dispersive velocity of ~320-330 m/s that was not observed for the class of events attributed to cryoseisms.

The temporal resolution of this study is limited due to the fact that data was recorded during specific intervals, rather than continuously (see Figure 12). This means that additional cryoseisms may have occurred under rapid cooling events that occurred during the field campaigns but for which no data was recorded. However, we can observe that the recording windows for which no cryoseisms were detected were associated with either temperatures that were too high or changing slowly in comparison to the periods when cryoseisms were detected. The frequency of cryoseisms was greater during the spring

compared to the autumn, probably owing to the increased progression of ground freezing in the aftermath of cold winter air temperatures. It is interesting to note that the highest frequency of cryoseisms was recorded the 2[nd] of May, 2019 during a period when the air temperature was rapidly increasing and following a sharp cold snap down from above-freezing temperatures three days prior. It is unclear whether these events are a delayed effect of the preceding cold snap, where the subsurface stress continues to increase for some time after the drop in air temperature, or if the events are caused by the sharp

temperature rise itself and associated with cracking driven by thermal expansion rather contraction. We also note that snow cover on the raised river terrace was thin or absent during the field campaigns, due to relatively low precipitation and strong prevailing winds. The lack of an insulating snow layer increases the plausibility of correlating air temperature at 5 m above





ground with cryoseismic events in the shallow subsurface and has been recognised as a necessary condition facilitating sufficiently rapid ground cooling to generate cryoseisms in previous studies (Barosh, 2000; Battaglia et al., 2016; Matsuoka et
al., 2018; Nikonov, 2010).

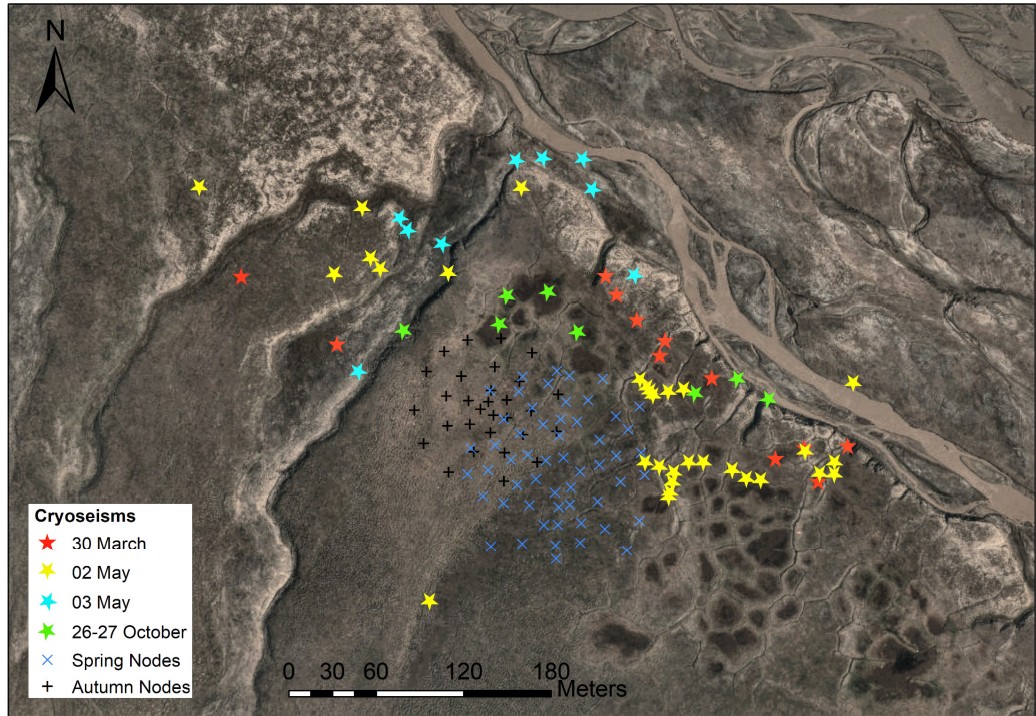

**Figure 11: Localised source positions (coloured stars) and receiver positions for spring (cross symbols) and autumn (plus symbols) field campaigns, background is a contrast enhanced version of an orthophoto © Norwegian Polar Institute (npolar.no).**



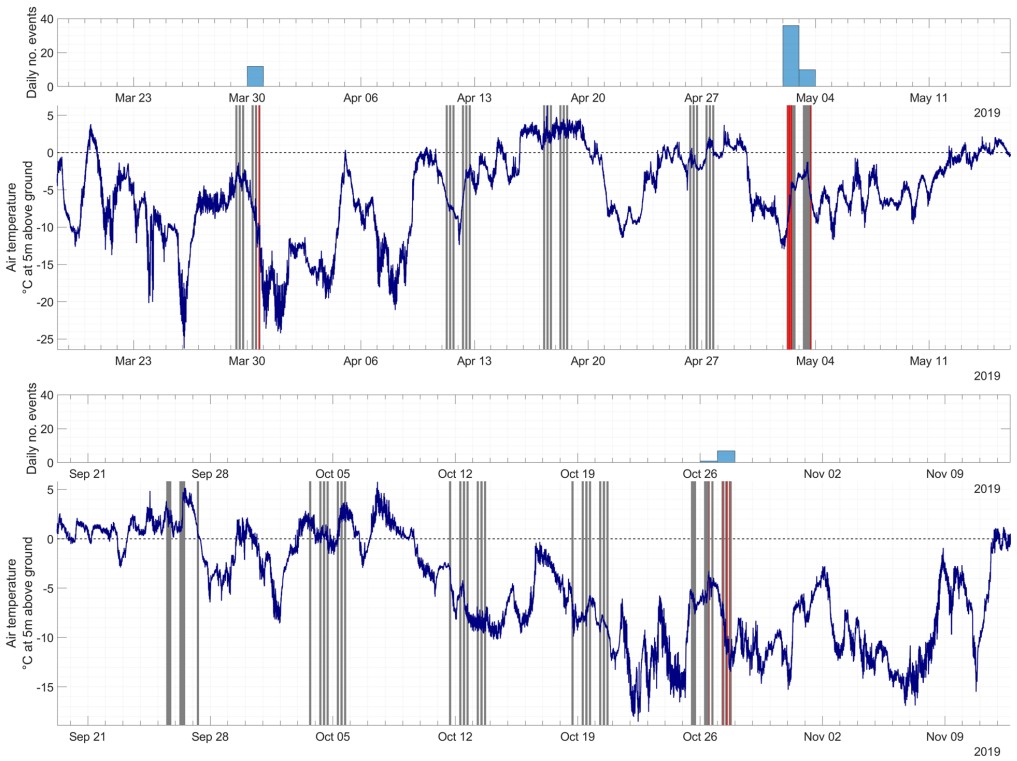

**Figure 12 – Temperature at 5 m above ground at nearby weather station in Adventdalen. Grey bars denote periods when passive seismic data was recorded and red bars denote periods where transient seismic events producing high quality dispersion spectra were detected.**

### 4.2    Dispersion images and temporal variation

Dispersion images from the isolated cryoseisms resemble the complex multimodal dispersion of Lamb-Rayleigh waves that is relatively well known from pavement studies (Ryden and Lowe, 2004; Ryden et al., 2004). This complexity emerges when a stiff high-velocity layer overlies a softer layer producing an inversion in the shear velocity profile and acting as a waveguide. Some examples of estimated dispersion spectra are shown in Figure 13 spanning both spring and autumn field campaigns. We observe that the number of wavemode branches imaged in the spring records was higher than in the autumn over the investigated range of frequencies. In Figure 14, we compare individual events from spring and autumn, and observe that the apparent dispersion curve is shifted towards lower velocities in the autumn and that transitions between successive modes are shifted to higher frequencies with larger spacing between modes. This trend is robust across the catalogue of cryoseisms giving well resolved dispersion images, as shown in Figure 15, displaying the time-frequency traced ridges of dispersion images





corresponding to multiple records from spring and autumn field campaigns. Matlab's built in routine "*tfridge*" was found to be effective for ridge tracing in this study.


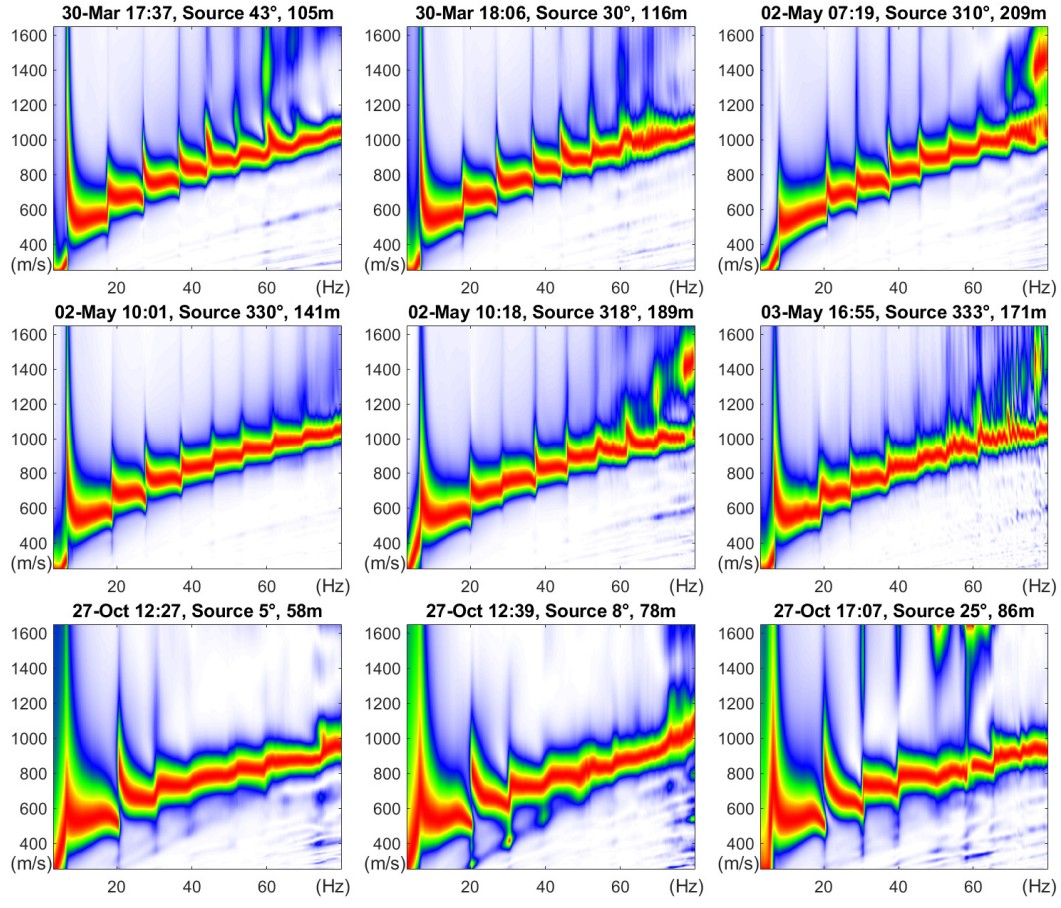

**Figure 13: Examples of dispersion spectra for selected cryoseismic events from spring (upper two rows) and autumn (bottom row) field campaigns, source azimuths are given as compass bearings and colour scaling is linear.**

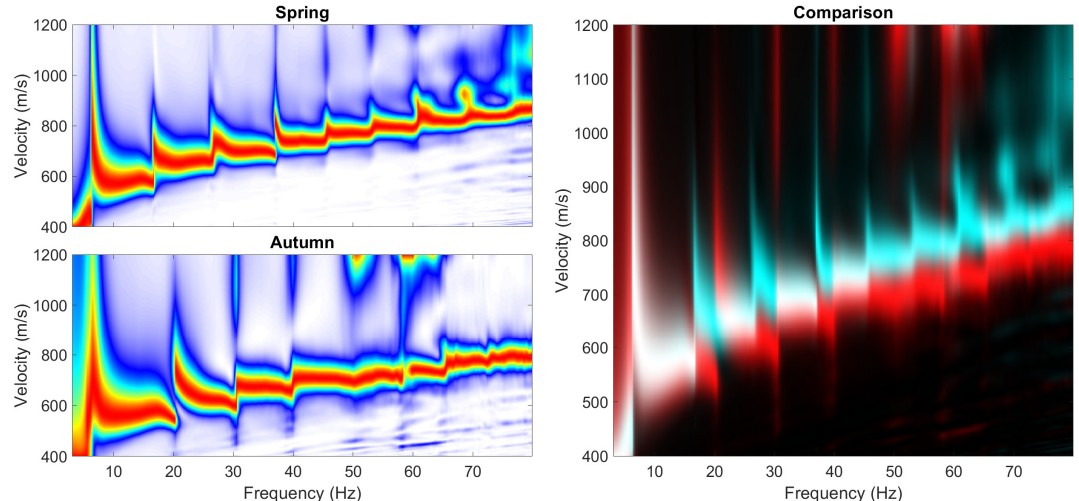


**Figure 14: Comparison of records from spring and autumn, in right panel spring record is shown in cyan and autumn record is shown in red, and areas where the two spectra overlap appear white.**

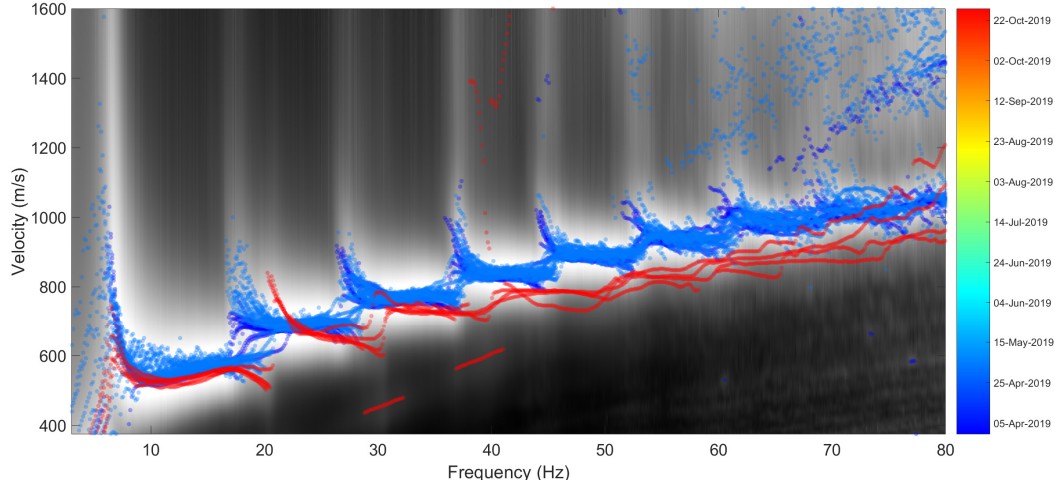


**Figure 15: Illustration of temporal variation. Grayscale background image shows the mean dispersion spectrum for all displayed transient events. Coloured circles denote time frequency ridges picked from individual dispersion spectra and coloured according to date of recording.**





### 4.3 Inferring subsurface structure from dispersion images

To further investigate what the structure of the dispersion images tells us about the subsurface permafrost structure and its
variation between spring and autumn field campaigns, we ran a series of theoretical models using the global matrix approach
discussed in sect. 3.4. These models were optimised manually by qualitatively fitting the resulting dispersion curves with
experimental dispersion images and manually adjusting the physical parameters to achieve a best possible fit. We focussed our
attention on the simplest possible models that give a good approximation of the experimental data, which in this case meant
two discrete layers with an inverse velocity profile sandwiched between infinite vacuum (above) and solid (below) half-spaces.
The physical properties of the best estimate models corresponding to spring and autumn conditions are listed in Table 1.

**Table 1: Physical properties of homogeneous, horizontally layered media used to calculate theoretical dispersion curves shown in Figure 16. The key feature of the model is a high velocity surface layer overlying a layer with lower velocity and high Poisson's ratio.**

| Layer | h (m) | | $V_s$ (m/s) | | $V_p$ (m/s) | | Poisson's Ratio | | $\rho$ |
|---|---|---|---|---|---|---|---|---|---|
| | Spring | Autumn | Spring | Autumn | Spring | Autumn | Spring | Autumn | (kg/m³) |
| 1 | 4.5 | 3.5 | 1700 | 1520 | 3180 | 3164 | 0.30 | 0.35 | 2000 |
| 2 | 31 | 29 | 500 | 525 | 1837 | 1929 | 0.46 | 0.46 | 2000 |
| halfspace | ∞ | ∞ | 2000 | 2000 | 3742 | 3742 | 0.30 | 0.30 | 2000 |

The primary property of the models that allows us to fit the experimental data is the high velocity of the uppermost part of the
ground, overlying relatively low velocity material beneath. We also see evidence that the Poisson's ratio in the low velocity
layer is relatively high (0.46), consistent with a softer material that transmits shear stress less effectively. The physical
manifestation of the high-velocity surface layer is likely to be the zone of elevated ground-ice content of ~4 m thickness
observed in the Adventdalen boreholes of Gilbert et al. (2018). This zone is rich in void filling ice, lenticular and massive solid
bodies of ice that macroscopically strengthen the dominantly loess sediments (Gilbert et al., 2018), leading to relatively high
shear wave velocity and a relatively low Poisson's ratio.

The low velocity layer may simply reflect the absence of these stiffening ice bodies and subsequently decreased shear strength
in the porous medium. However, the low velocity zone may also indicate the presence of unfrozen permafrost due to elevated
salinity, recalling that permafrost is defined simply as ground that remains below 0°C over at least two consecutive years, but
does not imply that the ground is in fact frozen. This interpretation is supported by the high Poisson's ratio of the lower layer,
since according to Skvortsov et al. (2014) a Poisson's ratio of 0.45-0.46 represents a threshold between frozen and unfrozen
states for water-saturated soils, irrespective of composition, temperature and salinity. Unfrozen saline permafrost has also been
interpreted in Adventdalen below the Holocene marine limit based on nuclear magnetic resonance and controlled source audio-



magnetotelluric data (Keating et al., 2018), which includes the study site. On balance, we conclude that unfrozen saline permafrost is the most likely explanation for the observed low velocity zone.

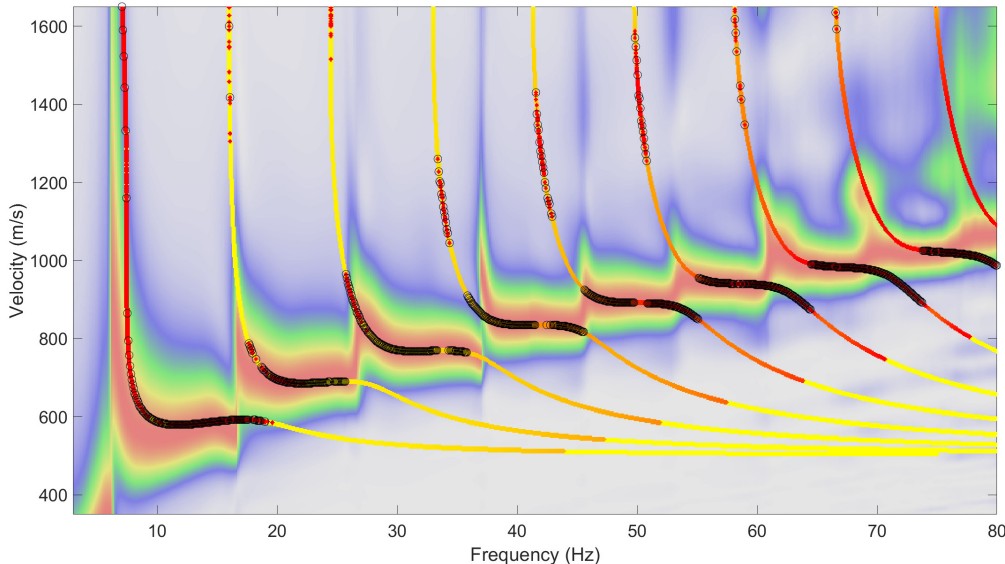

**Figure 16: Spring field campaign best qualitative fit theoretical dispersion curves, based on a simple 3 layer horizontal model (see**
**Table 1). Dispersion spectrum corresponds to event recorded 2-May 06:51, displayed with linear colour scaling.**

In Figure 16 we see that the theoretical dispersion curves fit the experimental data recorded in spring remarkably well, given our very simplistic layer model. Figure 17 illustrate that the fit between model and experimental data is somewhat poorer for the autumn, although good overall fit was still achieved. Reasons for this contrast may include that the cryoseisms were stronger in spring due to colder temperatures and a more advanced state of freezing leading to a more broadband source signal.

Alternately, the ground may have been more heterogeneous in the autumn, as indicated by interspersed ponds of unfrozen water and ice observed at the study site when deploying geophones in September compared with a relatively homogeneous frozen landscape with thin snow cover in March. This increased heterogeneity may affect the experimentally recorded events either via attenuation of the surface waves between source and receiver or by heterogeneities across the geophone array itself leading to decreased coherency.


It was difficult to fit the steep phase velocity gradients at the frequencies where the ground response transitions from one wave mode to another using our drastically simplified layer model (particularly noticeable for modes 3-6 in Figure 16). We have not



investigated this phenomenon in detail but hypothesise that some additional degree of freedom such as allowing for velocity gradients within layers may be required to improve this aspect of the fit.


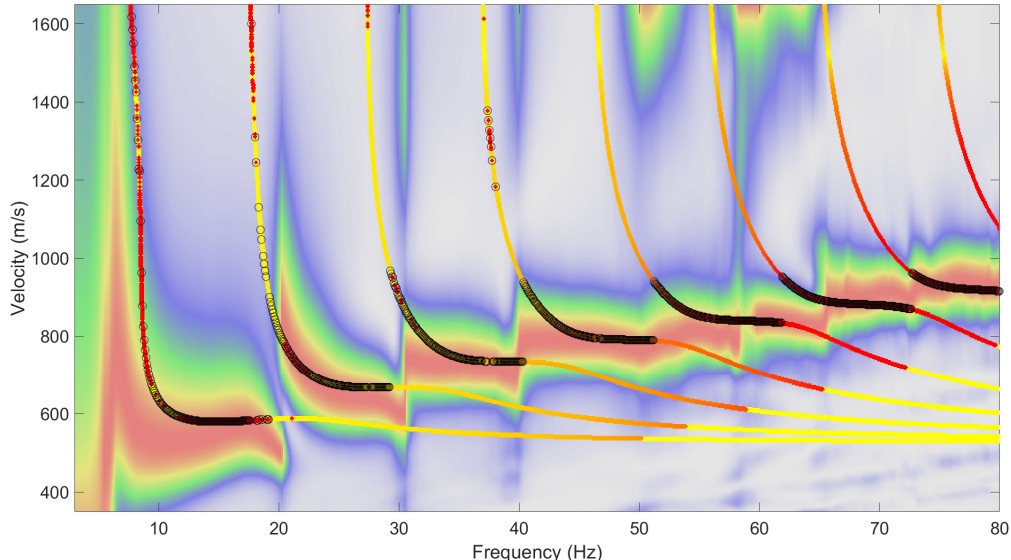

**Figure 17: Autumn field campaign best qualitative fit theoretical dispersion curves, based on a simple 3 layer horizontal model (see Table 1). Dispersion spectrum corresponds to event recorded 27-Oct 12:27, displayed with linear colour scaling.**

## 5  Conclusion

We present a methodology designed to isolate transient seismic events in passive records and thereby estimate their unknown source location and image their phase velocity dispersion. The spatial association of the source positions with a well-known frost polygon area along an elevated river-terrace in Adventdalen, together with temporal correlation with periods of rapidly changing air temperature, indicates that these events are likely cryoseisms. The phase velocity dispersion of these cryoseisms furthermore allows us to infer the subsurface structure of the permafrost and detect changes between seasons. A high-velocity

solid-frozen surface layer overlying a slower and softer layer leads to a complex multimodal dispersion pattern that is familiar from previous studies of pavements. The uppermost part of the permafrost appears to be measurably softer during the autumn than the spring, implying that this methodology may also have the potential to detect changes in an inter-annual monitoring context. A future field campaign recording continuously over an entire freeze season would, for example, give a more complete picture of the spatiotemporal occurrence of cryoseisms. Alternatively, our methodology could be applied for other locations

with suitable seismic sources, such as on or adjacent to glaciers.



## 6    Acknowledgments

This research is funded by the University of Tromsø - The Arctic University of Norway, by the ARCEx partners and by the Research Council of Norway through grant number 228107.

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
