# Peer review of "Passive seismic recording of cryoseisms in Adventdalen, Svalbard"

_The Cryosphere, 2020_

## Referee Comment (RC1) · Anonymous Referee #1 · 4 Aug 2020

I read this manuscript with pleasure: it is very well written and very detailed and therefore very clear. As a seismologist, I was interested to see that other surface-waves analysis and modelling tools were used in other communities. While the names and terminology were different, I could recognize familiar approaches.

I think the analysis presented in the paper is robust, and the data and results support the interpretations and conclusion. I only have minor comments to make to add to the discussion.

1) I feel that the events location method used in the paper is very similar to the Match Field Processing (MFP) described by Sergent et al. (2020) (adapted from previous sources). The advantage of the MFP is that its methodology is clearly defined both theoretically and practically. It is also quite adaptive with several tunings that can make

it very high resolution. I wonder why the authors did not try this approach with their arrays which are very well designed for that. Could you comment on that and perhaps compare the different methods?

2) In the aim of continuous long-term deployment to perform temporal monitoring of the permafrost, the fit "by hand" of the dispersion curves with their forward modelling in a 1D model seems inadequate. The best strategy, in this case, is to invert the dispersion curves to obtain the best fitting layered velocity model. There are many examples in the literature using this approach. However, it is quite rare to observe that many higher modes in data and existing inversion strategies might not be able to take all the modes into account. From your forward modelling strategy, would it be easy to design an inverse procedure? If yes, how would you do it? If no, what approach would you take? I think discussing this point would be a nice adjunction to the manuscript.

---

## Referee Comment (RC2) · Anonymous Referee #2 · 21 Aug 2020

This is a very well written and thorough presentation of research demonstrating the ability of using passive seismic methods to monitor seasonal changes in an actively changing permafrost environment. The analysis, particularly the source localization seems robust. The differences in Spring vs Autumn dispersion curves (Figures 14 and 15) is intriguing and at a quick glance suggests significant changes in the surface material properties, presumably related to the changing thermal environment, however on closer inspection it's more difficult for a reader to interpret precisely what is happening. The interpretation presented by the authors is of a thin (3.5-4.5 m) high velocity layer over a thicker (30 m) low velocity layer, which is quite thin compared to the presumed topography of the survey site and the source locations appear to have distinct regional variations. This needs to be accounted for in interpreting the differences between the

dispersion curves. A straightforward way to do this would be to see a direct comparison between one or two specific pairs of closely spaced individual events (Autumn, Spring) to see how much they vary with one another. This is especially important as the footprint of the recording nodes is slightly different between Spring and Autumn. It would also be helpful to see the actual model and it's range of variation plotted out.

I agree with the comment of referee #1 that I would typically attempt to invert the set of dispersion curves to obtain a layered velocity model result, if only to determine the range of models that fit the data. It's possible that this would illuminate features of interest and allow more predictive analyses. However, I don't consider this necessary for the acceptance of the paper itself.

---

## Author Comment (AC1) · 7 Sep 2020

Comment from Referee 1

"I feel that the events location method used in the paper is very similar to the Match Field Processing (MFP) described by Sergent et al. (2020) (adapted from previous sources). The advantage of the MFP is that its methodology is clearly defined both theoretically and practically. It is also quite adaptive with several tunings that can make it very high resolution. I wonder why the authors did not try this approach with their arrays which are very well designed for that. Could you comment on that and perhaps compare the different methods?"

Response:

This was a useful comment from the reviewer. We had not considered the MFP technique in this context initially, having been motivated to develop a method that is completely blind to specification of the velocity structure of the study site. We have now tested the specific MFP implementation described in Walter et. al. (2015) that is also referred to in Sergeant et al. (2020). In our implementation, we omit the ensemble averaging over multiple seismic noise windows since we are dealing with large amplitude, distinct microseismic events. Similarly to Walter et. al. (2015), we neglect amplitude information and match only the wave phase.

The additional constraint provided by specifying a model of the medium velocity means that the MFP method performs very well. Azimuth and range localisation was very reliable for highly idealised synthetic test data, when the medium velocity was accurately specified. We used the dominant A0 mode trend to specify the dispersion curve of the medium. Usefully, the medium velocity can also be estimated for real data via the simple azimuth scanning technique of Park et. al. (1998) to approximately locate the source azimuth and build a dispersion image. The MFP method that can then be used to estimate the source range and refine the source azimuth estimation. This approach may provide a way to overcome the challenge of relative offset invariance that occurs when the source is located far from the array and that limits the maximum range to which our trace sorting method is useful.

We have also tested the MFP method on our catalogue of microseismic events, where we find that source localisation is equivalent to our trace sorting method. The MFP method appears to be robust to the presence of random noise, but we still have the impression that our trace resorting method performs very well despite the challenges of receiver positioning uncertainty, non-stationary and non-uniform noise that we encounter in real data.

Actions for revised manuscript (1) The MFP source localisation results for our catalogue of microseismic events will be added to Figure 5. This will provide an additional benchmark that helps the reader set the source positioning results using our trace sort-

ing method in context. We include the revised figure also here as an attachment to this comment. (2) We will acknowledge throughout the text that our trace sorting method is an alternative to the existing MFP methodology that is also capable of resolving both azimuth and range of unknown seismic sources, under the additional constraint of a modelled medium velocity.
* * *
[Figure]

**Fig. 1.** Comparison of source localisation using Azimuth Scanning, (green crosses), Matched field processing (red triangles) and our trace re-sorting approach (blue circles)

---

## Author Comment (AC3) · 7 Sep 2020

Comment from Referee 2

"The differences in Spring vs Autumn dispersion curves (Figures 14 and 15) is intriguing and at a quick glance suggests significant changes in the surface material properties, presumably related to the changing thermal environment, however on closer inspection it's more difficult for a reader to interpret precisely what is happening. The interpretation presented by the authors is of a thin (3.5-4.5 m) high velocity layer over a thicker (30 m) low velocity layer, which is quite thin compared to the presumed topography of the survey site and the source locations appear to have distinct regional variations. This needs to be accounted for in interpreting the differences between the

dispersion curves. A straightforward way to do this would be to see a direct comparison between one or two specific pairs of closely spaced individual events (Autumn, Spring) to see how much they vary with one another. This is especially important as the footprint of the recording nodes is slightly different between Spring and Autumn. It would also be helpful to see the actual model and it's range of variation plotted out."

Response:

It is a good point that spatial variation in ground properties should be considered in addition to temporal variation and this has not been sufficiently discussed in the manuscript. It would not be unexpected that the true ground structure varies according to a complex spatiotemporal pattern. It is rather difficult to resolve these effects from our limited catalogue of microseismic events and we agree that the varying receiver footprint further complicates the interpretation of these effects. However, we are confident that the varying receiver footprint does not play a significant role for a laterally homogenous, horizontally layered medium (based on tests with synthetic test data). One might anticipate that a complex interplay between spatially varying ground properties and the slightly varying footprint of the receiver array could manifest as a false temporal signal. On the other hand, the dispersion images for spatially dispersed events over short time windows (24-48 hrs) are highly consistent, which points towards a ground structure that is (relatively) homogeneous in space. We therefore infer that the slightly varying geometry does not play a major role and that time varying ground structure best explains the observed variation in dispersion images.

As suggested by the reviewer, we have analysed subsets of events that cluster spatially while being dispersed temporally. In this case we still observe the same trend as illustrated in Figure 15, i.e., that there is significant variation between events that occur at different times of the year while events occurring around the same time are very consistent. As long as our argumentation holds that the varying receiver geometry doesn't play a significant role, we are still left with the conclusion that the ground structure is varying temporally.

We will update the manuscript to include a more nuanced discussion of spatial versus temporal effects and the varying receiver geometry. We feel that is not worth including additional figures to illustrate subsets of events that cluster in space but are dispersed in time (or vice versa) since the reader will be presented with essentially the same trend that is already illustrated in Figure 15. These figures are however included as attachments to this comment for reference.

Actions for revised manuscript:

(1) Include a more nuanced discussion of spatial versus temporal effects and the potentially confounding factor of varying receiver geometry. We will describe the variation we observe for sets of events that cluster in space and disperse in time (and vice versa) to make it clearer to the reader why we think that temporal variation in ground structure best explains the observed variation in dispersion images for our catalogue of microseismic events.

[Figure]

[Figure]

**Fig. 1.** Dispersion trends for a spatially clustered subset of microseismic events. We observe significant variation between spring and autumn events.

**Cryoseism cluster**
★ 30 March
★ 02 May
★ 03 May
★ 26-27 October
× Spring Nodes
+ Autumn Nodes

**Fig. 2.** Map view illustrating the source positions of the events from Fig. 1

[Figure]

**Fig. 3.** Temporally clustered events that are dispersed in space (see Fig. 11 in manuscript for locations) display a highly consistent dispersion pattern.

---

## Author Comment (AC4) · 7 Sep 2020

We would like to express our gratitude to both of the reviewers for their efforts in evaluating this manuscript. Their comments were both well directed and constructive. This review process will contribute positively to the overall quality of the article.

–––––––––––––––––––––––––––

---

## Author Response (AR1)

**Overview of reviewer comments requiring manuscript revision**

**Source localisation/match field processing**

**Comment from Referee 1**

*"I feel that the events location method used in the paper is very similar to the Match Field Processing (MFP) described by Sergent et al. (2020) (adapted from previous sources). The advantage of the MFP is that its methodology is clearly defined both theoretically and practically. It is also quite adaptive with several tunings that can make it very high resolution. I wonder why the authors did not try this approach with their arrays which are very well designed for that. Could you comment on that and perhaps compare the different methods?"*

**Author Response**

This was a useful comment from the reviewer. We had not considered the MFP technique in this context initially, having been motivated to develop a method that is completely blind to specification of the velocity structure of the study site. We have now tested the specific MFP implementation described in Walter et. al. (2015) that is also referred to in Sergeant et al. (2020). In our implementation, we omit the ensemble averaging over multiple seismic noise windows since we are dealing with large amplitude, distinct microseismic events. Similarly to Walter et. al. (2015), we neglect amplitude information and match only the wave phase.

The additional constraint provided by specifying a model of the medium velocity means that the MFP method performs very well. Azimuth and range localisation was very reliable for highly idealised synthetic test data, when the medium velocity was accurately specified. We used the dominant A0 mode trend to specify the dispersion curve of the medium. Usefully, the medium velocity can also be estimated for real data via the simple azimuth scanning technique of Park et. al. (1998) to approximately locate the source azimuth and build a dispersion image. The MFP method that can then be used to estimate the source range and refine the source azimuth estimation. This approach may provide a way to overcome the challenge of relative offset invariance that occurs when the source is located far from the array and that limits the maximum range to which our trace sorting method is useful.

We have also tested the MFP method on our catalogue of microseismic events, where we find that source localisation is equivalent to our trace sorting method. The MFP method appears to be robust to the presence of random noise, but we still have the impression that our trace resorting method performs very well despite the challenges of receiver positioning uncertainty, non-stationary and non-uniform noise that we encounter in real data.

**Manuscript revisions**

- The MFP source localisation results for our catalogue of microseismic events has been added to Figure 5. This will provide an additional benchmark that helps the reader set the source positioning results using our trace sorting method in context.
- We have softened the language around our source localisation method, given that it is one of several possible choices that appear to work equally well for our particular dataset.

**Inversion of dispersion curves/spectra**

**Comment from Referee 1**

*"In the aim of continuous long-term deployment to perform temporal monitoring of the permafrost, the fit "by hand" of the dispersion curves with their forward modelling in a 1D model seems inadequate. The best strategy, in this case, is to invert the dispersion curves to obtain the best fitting layered velocity model. There are many examples in the literature using this approach. However, it is quite rare to observe that many higher modes in data and existing inversion strategies might not be able to take all the modes into account. From your forward modelling strategy, would it be easy to design an inverse procedure? If yes, how would you do it? If no, what approach would you take?*

*I think discussing this point would be a nice adjunction to the manuscript."*

**Comment from Referee 2**

*"I agree with the comment of referee #1 that I would typically attempt to invert the set of dispersion curves to obtain a layered velocity model result, if only to determine the range of models that fit the data. It's possible that this would illuminate features of interest and allow more predictive analyses. However, I don't consider this necessary for the acceptance of the paper itself."*

**Author Response**

Within the context of a long-term monitoring application, it would be highly advantageous to develop an inversion scheme capable of robustly selecting the physical model(s) that fit the observed dispersion spectra. Implementing such an inversion scheme is a non-trivial undertaking due to the non-uniqueness and non-linearity of the problem, exacerbated by the complex multimodal dispersion structure that makes it difficult to optimise inversion parameters by exploiting partial derivatives. A valuable discussion of this topic is found in Ryden & Park, (2006). Because of this complexity, inversion of dispersion spectra fell outside the scope of the present study. However, a global optimisation technique such as the fast-simulated annealing approach implemented by Ryden & Park (2006) may provide a useful template that is likely applicable also for the type of data we have recorded. This method appears capable of dealing with the problems of cost-function local minima and correlated parameters assuming that the parameter perturbation size and "cooling schedule" of the simulated annealing can optimised to the specific set of dispersion spectra. We also see potential value in working further to understand the physical significance, from a wave-theoretical perspective, of some of the high-level structures observed in the dispersion spectra, such as the frequency spacing of mode transitions. Such features could also provide a means to measure some of the important physical properties of the system, in a potentially more transparent way than a multi-parameter inversion of the entire dispersion spectrum.

**Manuscript revisions**

- A brief discussion of the challenges of inversion and the relevant previous implementation of Ryden & Park, (2006) has been added to the discussion section of the manuscript.

**Interpreting variation between Spring & Autumn data**

**Comment from Referee 2**

*"The differences in Spring vs Autumn dispersion curves (Figures 14 and 15) is intriguing and at a quick glance suggests significant changes in the surface material properties, presumably related to the changing thermal environment, however on closer inspection it's more difficult for a reader to interpret precisely what is happening. The interpretation presented by the authors is of a thin (3.5-4.5 m) high velocity layer over a thicker (30 m) low velocity layer, which is quite thin compared to the presumed topography of the survey site and the source locations appear to have distinct regional variations. This needs to be accounted for in interpreting the differences between the dispersion curves. A straightforward way to do this would be to see a direct comparison between one or two specific pairs of closely spaced individual events (Autumn, Spring) to see how much they vary with one another. This is especially important as the footprint of the recording nodes is slightly different between Spring and Autumn. It would also be helpful to see the actual model and it's range of variation plotted out."*

**Author Response**

It is a good point that spatial variation in ground properties should be considered in addition to temporal variation and this has not been sufficiently discussed in the manuscript. It would not be unexpected that the true ground structure varies according to a complex spatiotemporal pattern. It is rather difficult to resolve these effects from our limited catalogue of microseismic events and we agree that the varying receiver footprint further complicates the interpretation of these effects. However, we are confident that the varying receiver footprint does not play a significant role for a laterally homogenous, horizontally layered medium (based on tests with synthetic test data). One might anticipate that a complex interplay between spatially varying ground properties and the slightly varying footprint of the receiver array could manifest as a false temporal signal. On the other hand, the dispersion images for spatially dispersed events over short time windows (24-48 hrs) are highly consistent (e.g. Figure x), which points towards a ground structure that is (relatively) homogeneous in space. We therefore infer that the slightly varying geometry does not play a major role and that time varying ground structure best explains the observed variation in dispersion images.

As suggested by the reviewer, we have analysed subsets of events that cluster spatially while being dispersed temporally (e.g. Figure y and Figure z). In this case we still observe the same trend as illustrated in Figure 15, i.e., that there is significant variation between events that occur at different times of the year while events occurring around the same time are very consistent. As long as our argumentation holds that the varying receiver geometry doesn't play a significant role, we are still left with the conclusion that the ground structure is varying temporally.

We will update the manuscript to include a more nuanced discussion of spatial versus temporal effects and the varying receiver geometry. We feel that is not worth including additional figures to illustrate subsets of events that cluster in space but are dispersed in time (or vice versa) since the reader will be presented with essentially the same trend that is already illustrated in Figure 15. However, we include the additional figures here for reference.

**Manuscript revisions**

- We have Included a short paragraph that justifies why we interpret the observed variation in dispersion images as a temporal change in ground structure as opposed change as a result of lateral inhomogeneity or varying receiver footprint.

Support figures only, these will not be included in the revised manuscript

[Figure]

*Figure x - Temporal cluster of events shows similar dispersion pattern*

[Figure]

*Figure y - Spatial cluster of events shows dispersion pattern that varies with time*

[Figure]

*Figure z – distribution of spatially clustered events.*

**General thanks**

We would like to express our gratitude to the reviewers for their efforts in evaluating this manuscript. Their comments were both well directed and constructive. This review process contributed positively to the overall quality of the article.

**List of changes in revised manuscript**

1. Figure 5 updated to include comparison with Matched Field Processing as suggested by reviewer.
2. Added definition of imaginary unit "j" in Eq. 1
3. Figure 9 – changed caption text to explicitly mention the true source position
4. Section 4.1 – broke up long sentence about other possible seismic sources
5. Figure 12 – increased font size of annotations, nonspecific "high quality dispersion spectra" changed to "multimodal dispersion spectra" in figure caption.
6. Section 4.3 – made reference to Keating et. al. (2018) more concise as it is already introduced in introduction.
7. Section 4.3 – added short paragraph about the complexity of inverting dispersion curves for ground structure which fell outside the scope of the present study.
8. Section 4.2 – added justification for interpretation that observed variation in dispersion images is due to temporal ground structure change, rather than spatial heterogeneity or receiver footprint.
9. Section 3.3.1 – Matched field processing described as an alternative method for source localisation that performs very similarly for our catalogue of microseismic events.
10. Section 3.3.1 – Velocity ambiguity check based on FK transform replaced by assigning minimum expected delay between receiver pairs. This change was brought about when re-evaluating the source localisation procedure following the matched-field processing comment from the reviewer and is a simpler and more efficient way to achieve the same task.
11. Figure 7 – simplified figure title and caption.
12. Minor correction/simplification of sentence structure.
13. Fixed small typographical issues in Reference list.

The following documents are attached:

1. Revised manuscript with markup showing all changes
2. "clean" revised manuscript

[revised manuscript text omitted]